# On the Fine-Grained Complexity of Empirical Risk Minimization: Kernel Methods and Neural Networks

**Arturs Backurs**
CSAIL
MIT
backurs@mit.edu

**Piotr Indyk**
CSAIL
MIT
indyk@mit.edu

**Ludwig Schmidt**
CSAIL
MIT
ludwigs@mit.edu

## Abstract

Empirical risk minimization (ERM) is ubiquitous in machine learning and under-lies most supervised learning methods. While there is a large body of work on algorithms for various ERM problems, the exact computational complexity of ERM is still not understood. We address this issue for multiple popular ERM problems including kernel SVMs, kernel ridge regression, and training the final layer of a neu-ral network. In particular, we give conditional hardness results for these problems based on complexity-theoretic assumptions such as the Strong Exponential Time Hypothesis. Under these assumptions, we show that there are no algorithms that solve the aforementioned ERM problems to high accuracy in sub-quadratic time. We also give similar hardness results for computing the gradient of the empirical loss, which is the main computational burden in many non-convex learning tasks.

## 1 Introduction

Empirical risk minimization (ERM) has been highly influential in modern machine learning [37]. ERM underpins many core results in statistical learning theory and is one of the main computational problems in the field. Several important methods such as support vector machines (SVM), boosting, and neural networks follow the ERM paradigm [34]. As a consequence, the algorithmic aspects of ERM have received a vast amount of attention over the past decades. This naturally motivates the following basic question:

*What are the computational limits for ERM algorithms?*

In this work, we address this question both in convex and non-convex settings. Convex ERM problems have been highly successful in a wide range of applications, giving rise to popular methods such as SVMs and logistic regression. Using tools from convex optimization, the resulting problems can be solved in polynomial time. However, the exact time complexity of many important ERM problems such as kernel SVMs is not yet well understood. As the size of data sets in machine learning continues to grow, this question is becoming increasingly important. For ERM problems with millions of high-dimensional examples, even quadratic time algorithms can become painfully slow (or expensive) to run.

Non-convex ERM problems have also attracted extensive research interest, e.g., in the context of deep neural networks. First order methods that follow the gradient of the empirical loss are not guaranteed to find the global minimizer in this setting. Nevertheless, variants of gradient descent are by far the most common method for training large neural networks. Here, the computational bottleneck is to compute a number of gradients, not necessarily to minimize the empirical loss globally. Although we

can compute gradients in polynomial time, the large number of parameters and examples in modern deep learning still makes this a considerable computational challenge.

Unfortunately, there are only few existing results concerning the exact time complexity of ERM or gradient computations. Since the problems have polynomial time algorithms, the classical machinery from complexity theory (such as NP hardness) is too coarse to apply. Oracle lower bounds from optimization offer useful guidance for convex ERM problems, but the results only hold for limited classes of algorithms. Moreover, they do not account for the *cost* of executing the oracle calls, as they simply lower bound their number. Overall, we do not know if common ERM problems allow for algorithms that compute a high-accuracy solution in sub-quadratic or even nearly-linear time for all instances.[1] Furthermore, we do not know if there are more efficient techniques for computing (mini-)batch gradients than simply treating each example in the batch independently.[2]

We address both questions for multiple well-studied ERM problems.

**Hardness of ERM.**   First, we give conditional hardness results for minimizing the empirical risk in several settings, including kernel SVMs, kernel ridge regression (KRR), and training the top layer of a neural network. Our results give evidence that no algorithms can solve these problems to high accuracy in strongly sub-quadratic time. Moreover, we provide similar conditional hardness results for kernel PCA. All of these methods are popular learning algorithms due to the expressiveness of the kernel or network embedding. Our results show that this expressiveness also leads to an expensive computational problem.

**Hardness of gradient computation in neural networks.**   Second, we address the complexity of computing a gradient for the empirical risk of a neural network. In particular, we give evidence that computing (or even approximating, up to polynomially large factors) the norm of the gradient of the top layer in a neural network takes time that is "rectangular". The time complexity cannot be significantly better than $O(n \cdot m)$, where $m$ is the number of examples and $n$ is the number of units in the network. Hence, there are no algorithms that compute batch gradients faster than handling each example individually, unless common complexity-theoretic assumptions fail.

Our hardness results for gradient computation apply to common activation functions such as ReLU or sigmoid units. We remark that for *polynomial* activation functions (for instance, studied in [24]), significantly faster algorithms *do exist*. Thus, our results can be seen as mapping the "efficiency landscape" of basic machine learning sub-routines. They distinguish between what is possible and (likely) impossible, suggesting further opportunities for improvement.

Our hardness results are based on recent advances in fine-grained complexity and build on conjectures such as the Strong Exponential Time Hypothesis (SETH) [23, 22, 38]. SETH concerns the classic satisfiability problem for formulas in Conjunctive Normal Form (CNF). Informally, the conjecture states that there is no algorithm for checking satisfiability of a formula with $n$ variables and $m$ clauses in time less than $O(c^n \cdot \text{poly}(m))$ for some $c < 2$.[3] While our results are conditional, SETH has been employed in many recent hardness results. Its plausibility stems from the fact that, despite 60 years of research on satisfiability algorithms, no such improvement has been discovered.

Our results hold for a significant range of the accuracy parameter. For kernel methods, our bounds hold for algorithms approximating the empirical risk up to a factor of $1 + \varepsilon$, for $\log(1/\varepsilon) = \omega(\log^2 n)$. Thus, they provide conditional quadratic lower bounds for algorithms with, say, a $\log 1/\varepsilon$ runtime dependence on the approximation error $\varepsilon$. A (doubly) logarithmic dependence on $1/\varepsilon$ is generally seen as the ideal rate of convergence in optimization, and algorithms with this property have been studied extensively in the machine learning community (cf. [12].). At the same time, approximate

solutions to ERM problems can be sufficient for good generalization in learning tasks. Indeed, stochastic gradient descent (SGD) is often advocated as an efficient learning algorithm despite its polynomial dependence on $1/\varepsilon$ in the optimization error [35, 15]. Our results support this viewpoint since SGD sidesteps the quadratic time complexity of our lower bounds.

For other problems, our assumptions about the accuracy parameter are less stringent. In particular, for training the top layer of the neural network, we only need to assume that $\varepsilon \approx 1/n$. Finally, our lower bounds for approximating the norm of the gradient in neural networks hold even if $\varepsilon = n^{O(1)}$, i.e., for *polynomial* approximation factors (or alternatively, a constant additive factor for ReLU and sigmoid activation functions).

Finally, we note that our results do not rule out algorithms that achieve a sub-quadratic running time for well-behaved instances, e.g., instances with low-dimensional structure. Indeed, many such approaches have been investigated in the literature, for instance the Nyström method or random features for kernel problems [40, 30]. Our results offer an explanation for the wide variety of techniques. The lower bounds are evidence that there is no "silver bullet" algorithm for solving the aforementioned ERM problems in sub-quadratic time, to high accuracy, and for all instances.

## 2  Background

**Fine-grained complexity.**    We obtain our conditional hardness results via reductions from two well-studied problems: *Orthogonal Vectors* and *Bichromatic Hamming Close Pair*.

**Definition 1** (Orthogonal Vectors problem (OVP))**.** *Given two sets $A = \{a_1, \ldots, a_n\} \subseteq \{0,1\}^d$ and $B = \{b_1, \ldots, b_n\} \subseteq \{0,1\}^d$ of $n$ binary vectors, decide if there exists a pair $a \in A$ and $b \in B$ such that $a^{\mathsf{T}} b = 0$.*

For OVP, we can assume without loss of generality that all vectors in $B$ have the same number of 1s. This can be achieved by appending $d$ entries to every $b_i$ and setting the necessary number of them to 1 and the rest to 0. We then append $d$ entries to every $a_i$ and set all of them to 0.

**Definition 2** (Bichromatic Hamming Close Pair (BHCP) problem)**.** *Given two sets $A = \{a_1, \ldots, a_n\} \subseteq \{0,1\}^d$ and $B = \{b_1, \ldots, b_n\} \subseteq \{0,1\}^d$ of $n$ binary vectors and an integer $t \in \{2, \ldots, d\}$, decide if there exists a pair $a \in A$ and $b \in B$ such that the number of coordinates in which they differ is less than $t$ (formally, Hamming$(a, b) := ||a - b||_1 < t$). If there is such a pair $(a, b)$, we call it a* close pair*.*

It is known that both OVP and BHCP require almost quadratic time (i.e., $n^{2-o(1)}$) for any $d = \omega(\log n)$ assuming SETH [5].[4] Furthermore, if we allow the sizes $|A| = n$ and $|B| = m$ to be different, both problems require $(nm)^{1-o(1)}$ time assuming SETH, as long as $m = n^\alpha$ for some constant $\alpha \in (0, 1)$ [17]. Our proofs will proceed by embedding OVP and BHCP instances into ERM problems. Such a reduction then implies that the ERM problem requires almost quadratic time if the SETH is true. If we could solve the ERM problem faster, we would also obtain a faster algorithm for the satisfiability problem.

## 3  Our contributions

### 3.1  Kernel ERM problems

We provide hardness results for multiple kernel problems. In the following, let $x_1, \ldots, x_n \in \mathbb{R}^d$ be the $n$ input vectors, where $d = \omega(\log n)$. We use $y_1, \ldots, y_n \in \mathbb{R}$ as $n$ labels or target values. Finally, let $k(x, x')$ denote a kernel function and let $K \in \mathbb{R}^{n \times n}$ be the corresponding kernel matrix, defined as $K_{i,j} := k(x_i, x_j)$ [33]. Concretely, we focus on the Gaussian kernel $k(x, x') := \exp\left(-C\|x - x'\|_2^2\right)$ for some $C > 0$. We note that our results can be generalized to any kernel with exponential tail.

**Kernel SVM.** For simplicity, we present our result for hard-margin SVMs without bias terms. This gives the following optimization problem.

**Definition 3** (Hard-margin SVM). *A (primal) hard-margin SVM is an optimization problem of the following form:*

$$\underset{\alpha_1,\ldots,\alpha_n \geq 0}{minimize} \quad \frac{1}{2} \sum_{i,j=1}^{n} \alpha_i \, \alpha_j \, y_i \, y_j \, k(x_i, x_j) \tag{1}$$

$$subject\ to \quad y_i f(x_i) \geq 1, \ \ i = 1, \ldots, n,$$

*where* $f(x) := \sum_{i=1}^{n} \alpha_i y_i k(x_i, x)$.

The following theorem is our main result for SVMs, described in more detail in Section 4. In Sections B, C, and D of the supplementary material we provide similar hardness results for other common SVM variants, including the soft-margin version.

**Theorem 4.** *Let $k(a, a')$ be the Gaussian kernel with $C = 100 \log n$ and let $\varepsilon = \exp(-\omega(\log^2 n))$. Then approximating the optimal value of Equation* (1) *within a multiplicative factor $1 + \varepsilon$ requires almost quadratic time assuming SETH.*

**Kernel Ridge Regression.** Next we consider Kernel Ridge Regression, which is formally defined as follows.

**Definition 5** (Kernel ridge regression). *Given a real value $\lambda \geq 0$, the goal of kernel ridge regression is to output*

$$\arg \min_{\alpha \in \mathbb{R}^n} \frac{1}{2} \|y - K\alpha\|_2^2 + \frac{\lambda}{2} \alpha^\mathsf{T} K\alpha.$$

This problem is equivalent to computing the vector $(K + \lambda I)^{-1}y$. We focus on the special case where $\lambda = 0$ and the vector $y$ has all equal entries $y_1 = \ldots = y_n = 1$. In this case, the entrywise sum of $K^{-1}y$ is equal to the sum of the entries in $K^{-1}$. Thus, we show hardness for computing the latter quantity (see Section F in the supplementary material for the proof).

**Theorem 6.** *Let $k(a, a')$ be the Gaussian kernel for any parameter $C = \omega(\log n)$ and let $\varepsilon = \exp(-\omega(\log^2 n))$. Then computing the sum of the entries in $K^{-1}$ up to a multiplicative factor of $1 + \varepsilon$ requires almost quadratic time assuming SETH.*

**Kernel PCA.** Finally, we turn to the Kernel PCA problem, which we define as follows [26].

**Definition 7** (Kernel Principal Component Analysis (PCA)). *Let $1_n$ be an $n \times n$ matrix where each entry takes value $1/n$, and define $K' := (I - 1_n)K(I - 1_n)$. The goal of the kernel PCA problem is to output the $n$ eigenvalues of the matrix $K'$.*

In the above definition, the output only consists of the eigenvalues, not the eigenvectors. This is because computing all $n$ eigenvectors trivially takes at least quadratic time since the output itself has quadratic size. Our hardness proof applies to the potentially simpler problem where only the eigenvalues are desired. Specifically, we show that computing *the sum* of the eigenvalues (i.e., the *trace* of the matrix) is hard. See Section E in the supplementary material for the proof.

**Theorem 8.** *Let $k(a, a')$ be the Gaussian kernel with $C = 100 \log n$ and let $\varepsilon = \exp(-\omega(\log^2 n))$. Then approximating the sum of the eigenvalues of $K' = (I - 1_n)K(I - 1_n)$ within a multiplicative factor of $1 + \varepsilon$ requires almost quadratic time assuming SETH.*

We note that the argument in the proof shows that even approximating the sum of the entries of $K$ is hard. This provides an evidence of hardness of the *kernel density estimation* problem for Gaussian kernels, complementing recent upper bounds of [20].

## 3.2 Neural network ERM problems

We now consider neural networks. We focus on the problem of optimizing the top layer while keeping lower layers unchanged. An instance of this problem is transfer learning with large networks that would take a long time and many examples to train from scratch [31]. We consider neural networks of depth 2, with the sigmoid or ReLU activation function. Our hardness result holds for a more general class of "nice" activation functions $S$ as described later (see Definition 12).

Given $n$ weight vectors $w_1, \ldots, w_n \in \mathbb{R}^d$ and $n$ weights $\alpha_1, \ldots, \alpha_n \in \mathbb{R}$, consider the function $f : \mathbb{R}^d \to \mathbb{R}$ using a non-linearity $S : \mathbb{R} \to \mathbb{R}$:

$$f(u) := \sum_{j=1}^{n} \alpha_j \cdot S(u^\mathsf{T} w_j) .$$

This function can be implemented as a neural net that has $d$ inputs, $n$ nonlinear activations (units), and one linear output.

To complete the ERM problem, we also require a loss function. Our hardness results hold for a large class of "nice" loss functions, which includes the hinge loss and the logistic loss.[5] Given a nice loss function and $m$ input vectors $a_1, \ldots, a_m \in \mathbb{R}^d$ with corresponding labels $y_i$, we consider the following problem:

$$\underset{\alpha_1, \ldots, \alpha_n \in \mathbb{R}}{\text{minimize}} \quad \sum_{i=1}^{m} \text{loss}(y_i, f(u_i)). \tag{2}$$

Our main result is captured by the following theorem (see Section 5 for the proof). For simplicity, we set $m = n$.

**Theorem 9.** *For any $d = \omega(\log n)$, approximating the optimal value in Equation* (2) *up to a multiplicative factor of $1 + \frac{1}{4n}$ requires almost quadratic time assuming SETH.*

### 3.3 Hardness of gradient computation

Finally, we consider the problem of computing the gradient of the loss function for a given set of examples. We focus on the network architecture from the previous section. Formally, we obtain the following result:

**Theorem 10.** *Consider the empirical risk in Equation* (2) *under the following assumptions: (i) The function $f$ is represented by a neural network with $n$ units, $n \cdot d$ parameters, and the ReLU activation function. (ii) We have $d = \omega(\log n)$. (iii) The loss function is the logistic loss or hinge loss. Then approximating the $\ell_p$-norm (for any $p \geq 1$) of the gradient of the empirical risk for $m$ examples within a multiplicative factor of $n^C$ for any constant $C > 0$ takes at least $O\left((nm)^{1-o(1)}\right)$ time assuming SETH.*

See Section 6 for the proof. We also prove a similar statement for the sigmoid activation function. At the same time, we remark that for *polynomial* activation functions, significantly faster algorithms do exist, using the polynomial lifting argument. Specifically, for the polynomial activation function of the form $x^r$ for some integer $r \geq 2$, all gradients can be computed in $O((n+m)d^r)$ time. Note that the running time of the standard backpropagation algorithm is $O(dnm)$ for networks with this architecture. Thus one can improve over backpropagation for a non-trivial range of parameters, especially for quadratic activation function when $r = 2$. See Section H in the supplementary material for more details.

### 3.4 Related work

Recent work has demonstrated conditional quadratic hardness results for many combinatorial optimization problems over graphs and sequences. These results include computing diameter in sparse graphs [32, 21], Local Alignment [2], Fréchet distance [16], Edit Distance [13], Longest Common Subsequence, and Dynamic Time Warping [1, 17]. In the machine learning literature, [14] recently showed a tight lower bound for the problem of inferring the most likely path in a Hidden Markov Model, matching the upper bound achieved by the Viterbi algorithm [39]. As in our paper, the SETH and related assumptions underlie these lower bounds. To the best of our knowledge, our paper is the first application of this methodology to *continuous* (as opposed to combinatorial) optimization problems.

There is a long line of work on the oracle complexity of optimization problems, going back to [28]. We refer the reader to [29] for these classical results. The oracle complexity of ERM problems is still

subject of active research, e.g., see [3, 19, 41, 9, 10]. The work closest to ours is [19], which gives quadratic time lower bounds for ERM algorithms that access the kernel matrix through an evaluation oracle or a low-rank approximation.

The oracle results are fundamentally different from the lower bounds presented in our paper. Oracle lower bounds are typically unconditional, but inherently apply only to a limited class of algorithms due to their information-theoretic nature. Moreover, they do not account for the *cost* of executing the oracle calls, as they merely lower bound their number. In contrast, our results are conditional (based on the SETH and related assumptions), but apply to *any* algorithm and account for the *total* computational cost. This significantly broadens the reach of our results. We show that the hardness is not due to the oracle abstraction but instead inherent in the computational problem.

## 4 Overview of the hardness proof for kernel SVMs

Let $A = \{a_1, \ldots, a_n\} \subseteq \{0, 1\}^d$ and $B = \{b_1, \ldots, b_n\} \subseteq \{0, 1\}^d$ be the two sets of binary vectors from a BHCP instance with $d = \omega(\log n)$. Our goal is to determine whether there is a close pair of vectors. We show how to solve this BHCP instance by reducing it to *three* computations of SVM, defined as follows:

1. We take the first set $A$ of binary vectors, assign label 1 to all vectors, and solve the corresponding SVM on the $n$ vectors:

$$\begin{aligned} \operatorname*{minimize}_{\alpha_1, \ldots, \alpha_n \geq 0} \quad & \frac{1}{2} \sum_{i,j=1}^{n} \alpha_i \alpha_j k(a_i, a_j) \\ \text{subject to} \quad & \sum_{j=1}^{n} \alpha_j k(a_i, a_j) \geq 1, \quad i = 1, \ldots, n. \end{aligned} \tag{3}$$

   Note that we do not have $y_i$ in the expressions because all labels are 1.

2. We take the second set $B$ of binary vectors, assign label $-1$ to all vectors, and solve the corresponding SVM on the $n$ vectors:

$$\begin{aligned} \operatorname*{minimize}_{\beta_1, \ldots, \beta_n \geq 0} \quad & \frac{1}{2} \sum_{i,j=1}^{n} \beta_i \beta_j k(b_i, b_j) \\ \text{subject to} \quad & -\sum_{j=1}^{n} \beta_j k(b_i, b_j) \leq -1, \quad i = 1, \ldots, n. \end{aligned} \tag{4}$$

3. We take both sets $A$ and $B$ of binary vectors, assign label 1 to all vectors from the first set $A$ and label $-1$ to all vectors from the second set $B$. We then solve the corresponding SVM on the $2n$ vectors:

$$\begin{aligned} \operatorname*{minimize}_{\substack{\alpha_1, \ldots, \alpha_n \geq 0 \\ \beta_1, \ldots, \beta_n \geq 0}} \quad & \frac{1}{2} \sum_{i,j=1}^{n} \alpha_i \alpha_j k(a_i, a_j) + \frac{1}{2} \sum_{i,j=1}^{n} \beta_i \beta_j k(b_i, b_j) - \sum_{i,j=1}^{n} \alpha_i \beta_j k(a_i, b_j) \\ \text{subject to} \quad & \sum_{j=1}^{n} \alpha_j k(a_i, a_j) - \sum_{j=1}^{n} \beta_j k(a_i, b_j) \geq 1, \quad i = 1, \ldots, n, \\ & -\sum_{j=1}^{n} \beta_j k(b_i, b_j) + \sum_{j=1}^{n} \alpha_j k(b_i, a_j) \leq -1, \quad i = 1, \ldots, n. \end{aligned} \tag{5}$$

**Intuition behind the construction.** To show a reduction from the BHCP problem to SVM computation, we have to consider two cases:

- The YES case of the BHCP problem when there are two vectors that are close in Hamming distance. That is, there exist $a_i \in A$ and $b_j \in B$ such that $\text{Hamming}(a_i, b_j) < t$.
- The NO case of the BHCP problem when there is no close pair of vectors. That is, for all $a_i \in A$ and $b_j \in B$, we have $\text{Hamming}(a_i, b_j) \geq t$.

We show that we can distinguish between these two cases by comparing the objective value of the first two SVM instances above to the objective value of the third.

**Intuition for the NO case.** We have $\text{Hamming}(a_i, b_j) \geq t$ for all $a_i \in A$ and $b_j \in B$. The Gaussian kernel then gives the inequality

$$k(a_i, b_j) = \exp(-100 \log n \cdot \|a_i - b_j\|_2^2) \leq \exp(-100 \log n \cdot t)$$

for all $a_i \in A$ and $b_j \in B$. This means that the value $k(a_i, b_j)$ is very small. For simplicity, assume that it is equal to 0, i.e., $k(a_i, b_j) = 0$ for all $a_i \in A$ and $b_j \in B$.

Consider the third SVM (5). It contains three terms involving $k(a_i, b_j)$: the third term in the objective function, the second term in the inequalities of the first type, and the second term in the inequalities of the second type. We assumed that these terms are equal to 0 and we observe that the rest of the third SVM is equal to the sum of the first SVM (3) and the second SVM (4). Thus we expect that the optimal value of the third SVM is approximately equal to the sum of the optimal values of the first and the second SVMs. If we denote the optimal value of the first SVM (3) by $\text{value}(A)$, the optimal value of the second SVM (4) by $\text{value}(B)$, and the optimal value of the third SVM (5) by $\text{value}(A, B)$, then we can express our intuition in terms of the approximate equality

$$\text{value}(A, B) \approx \text{value}(A) + \text{value}(B) \ .$$

**Intuition for the YES case.** In this case, there is a close pair of vectors $a_i \in A$ and $b_j \in B$ such that $\text{Hamming}(a_i, b_j) \leq t - 1$. Since we are using the Gaussian kernel we have the following inequality for this pair of vectors:

$$k(a_i, b_j) = \exp(-100 \log n \cdot \|a_i - b_j\|_2^2) \geq \exp(-100 \log n \cdot (t - 1)) \ .$$

We therefore have a large summand in each of the three terms from the above discussion. Thus the three terms do not (approximately) disappear and there is no reason for us to expect that the approximate equality holds. We can thus expect

$$\text{value}(A, B) \not\approx \text{value}(A) + \text{value}(B) \ .$$

Thus, by computing $\text{value}(A, B)$ and comparing it to $\text{value}(A) + \text{value}(B)$ we can distinguish between the YES and NO instances of BHCP. This completes the reduction. The full proofs are given in Section B of the supplementary material.

## 5 Overview of the hardness proof for training the final layer of a neural network

We start by formally defining the class of "nice" loss functions and "nice" activation functions.

**Definition 11.** *For a label $y \in \{-1, 1\}$ and a prediction $w \in \mathbb{R}$, we call the loss function $loss(y, w) : \{-1, 1\} \times \mathbb{R} \to \mathbb{R}_{\geq 0}$* nice *if the following three properties hold:*

- *$loss(y, w) = l(yw)$ for some convex function $l : \mathbb{R} \to \mathbb{R}_{\geq 0}$.*

- *For some sufficiently large constant $K > 0$, we have that (i) $l(x) \leq o(1)$ for all $x \geq n^K$, (ii) $l(x) \geq \omega(n)$ for all $x \leq -n^K$, and (iii) $l(x) = l(0) \pm o(1/n)$ for all $x \in \pm O(n^{-K})$.*

- *$l(0) > 0$ is some constant strictly larger than $0$.*

We note that the hinge loss function $loss(y, x) = \max(0, 1 - y \cdot x)$ and the logistic loss function $loss(y, x) = \frac{1}{\ln 2} \ln(1 + e^{-y \cdot x})$ are nice loss functions according to the above definition.

**Definition 12.** *A non-decreasing* activation functions *$S : \mathbb{R} \to \mathbb{R}_{\geq 0}$ is "nice" if it satisfies the following property: for all sufficiently large constants $T > 0$ there exist $v_0 > v_1 > v_2$ such that $S(v_0) = \Theta(1)$, $S(v_1) = 1/n^T$, $S(v_2) = 1/n^{\omega(1)}$ and $v_1 = (v_0 + v_2)/2$.*

The ReLU activation $S(z) = \max(0, z)$ satisfies these properties since we can choose $v_0 = 1$, $v_1 = 1/n^T$, and $v_2 = -1 + 2/n^T$. For the sigmoid function $S(z) = \frac{1}{1+e^{-z}}$, we can choose

$v_1 = -\log(n^T - 1)$, $v_0 = v_1 + C$, and $v_2 = v_1 - C$ for some $C = \omega(\log n)$. In the rest of the proof we set $T = 1000K$, where $K$ is the constant from Definition 11.

We now describe the proof of Theorem 9. We use the notation $\alpha := (\alpha_1, \ldots, \alpha_n)^\mathsf{T}$. Invoking the first property from Definition 11, we observe that the optimization problem (2) is equivalent to the following optimization problem:

$$\underset{\alpha \in \mathbb{R}^n}{\text{minimize}} \quad \sum_{i=1}^{m} l(y_i \cdot (M\alpha)_i), \tag{6}$$

where $M \in \mathbb{R}^{m \times n}$ is the matrix defined as $M_{i,j} := S(u_i^\mathsf{T} w_j)$ for $i = 1, \ldots, m$ and $j = 1, \ldots n$. For the rest of the section we will use $m = \Theta(n)$.[6]

Let $A = \{a_1, \ldots, a_n\} \subseteq \{0, 1\}^d$ and $B = \{b_1, \ldots, b_n\} \subseteq \{0, 1\}^d$ with $d = \omega(\log n)$ be the input to the Orthogonal Vectors problem. To show hardness we define a matrix $M$ as a vertical concatenation of 3 smaller matrices: $M_1$, $M_2$ and $M_2$ (repeated). Both matrices $M_1, M_2 \in \mathbb{R}^{n \times n}$ are of size $n \times n$. Thus the number of rows of $M$ (equivalently, the number of training examples) is $m = 3n$.

**Reduction overview.** We select the input examples and weights so that the matrices $M_1$ and $M_2$, have the following properties:

- $M_1$: if two vectors $a_i$ and $b_j$ are orthogonal, then the corresponding entry $(M_1)_{i,j} = S(v_0) = \Theta(1)$ and otherwise $(M_1)_{i,j} \approx 0$.[7]

- $M_2$: $(M_2)_{i,i} = S(v_1) = 1/n^{1000K}$ and $(M_2)_{i,j} \approx 0$ for all $i \neq j$

To complete the description of the optimization problem (6), we assign labels to the inputs corresponding to the rows of the matrix $M$. We assign label 1 to all inputs corresponding to rows of the matrix $M_1$ and the first copy of the matrix $M_2$. We assign label $-1$ to all remaining rows of the matrix $M$ corresponding to the second copy of matrix $M_2$.

The proof of the theorem is completed by the following two lemmas. See Section G in the supplementary material for the proofs.

**Lemma 13.** *If there is a pair of orthogonal vectors, then the optimal value of* (6) *is upper bounded by* $(3n - 1) \cdot l(0) + o(1)$.

**Lemma 14.** *If there is no pair of orthogonal vectors, then the optimal value of* (6) *is lower bounded by* $3n \cdot l(0) - o(1)$.

## 6 Hardness proof for gradient computation

Finally, we consider the problem of computing the gradient of the loss function for a given set of examples. We focus on the network architecture as in the previous section. Specifically, let $F_{\alpha,B}(a) := \sum_{j=1}^{n} \alpha_j S(a, b_j)$ be the output of a neural net with activation function $S$, where: (1) $a$ is an input vector from the set $A := \{a_1, \ldots, a_m\} \subseteq \{0, 1\}^d$; (2) $B := \{b_1, \ldots, b_n\} \subseteq \{0, 1\}^d$ is a set of binary vectors; (3) $\alpha = \{\alpha_1, \ldots, \alpha_n\}^T \in \mathbb{R}^n$ is an $n$-dimensional real-valued vector. We first prove the following lemma.

**Lemma 15.** *For some loss function* $l : \mathbb{R} \to \mathbb{R}$, *let* $l(F_{\alpha,B}(a))$ *be the loss for input* $a$ *when the label of the input* $a$ *is* $+1$. *Consider the gradient of the total loss* $l_{\alpha,A,B} := \sum_{a \in A} l(F_{\alpha,B}(a))$ *at* $\alpha_1 = \ldots = \alpha_n = 0$ *with respect to* $\alpha_1, \ldots, \alpha_n$. *The sum of the entries of the gradient is equal to* $l'(0) \cdot \sum_{a \in A, b \in B} S(a, b)$, *where* $l'(0)$ *is the derivative of the loss function* $l$ *at* 0.

For the hinge loss function, we have that the loss function is $l(x) = \max(0, 1 - x)$ if the label is $+1$. Thus, $l'(0) = -1$. For the logistic loss function, we have that the loss function is $l(x) = \frac{1}{\ln 2} \ln(1 + e^{-x})$ if the label is $+1$. Thus, $l'(0) = -\frac{1}{2\ln 2}$ in this case.

*Proof of Theorem 10.* Since all $\ell_p$-norms are within a polynomial factor, it suffices to show the statement for $\ell_1$-norm.

We set $S(a, b) := \max(0, 1 - 2a^\mathsf{T}b)$. Using Lemma 15, we get that the $\ell_1$-norm of the gradient of the total loss function is equal to $|l'(0)| \cdot \sum_{a \in A, b \in B} 1_{a^\mathsf{T}b=0}$. Since $l'(0) \neq 0$, this reduces OV to the gradient computation problem. Note that if there is no orthogonal pair, then the $\ell_1$-norm is $0$ and otherwise it is a constant strictly greater than $0$. Thus approximating the $\ell_1$-norm within any finite factor allows us to distinguish the cases. □

See Section H in the supplementary material for other results.

# 7 Conclusions

We have shown that a range of kernel problems require quadratic time for obtaining a high accuracy solution unless the Strong Exponential Time Hypothesis is false. These problems include variants of kernel SVM, kernel ridge regression, and kernel PCA. We also gave a similar hardness result for training the final layer of a depth-2 neural network. This result is general and applies to multiple loss and activation functions. Finally, we proved that computing the empirical loss gradient for such networks takes time that is essentially "rectangular", i.e., proportional to the product of the network size and the number of examples.

We note that our quadratic (rectangular) hardness results hold for *general* inputs. There is a long line of research on algorithms for kernel problems with running times depending on various input parameters, such as its statistical dimension [42], degrees of freedom [11] or effective dimensionality [27]. It would be interesting to establish lower bounds on the complexity of kernel problems as a function of the aforementioned input parameters.

Our quadratic hardness results for kernel problems apply to kernels with exponential tails. A natural question is whether similar results can be obtained for "heavy-tailed" kernels, e.g., the Cauchy kernel. We note that similar results for the linear kernel do not seem achievable using our techniques.[8]

Several of our results are obtained by a reduction from the (exact) Bichromatic Hamming Closest Pair problem or the Orthogonal Vectors problem. This demonstrates a strong connection between kernel methods and similarity search, and suggests that perhaps a reverse reduction is also possible. Such a reduction could potentially lead to faster approximate algorithms for kernel methods: although the exact closest pair problem has no known sub-quadratic solution, efficient and practical sub-quadratic time algorithms for the approximate version of the problem exist (see e.g., [6, 36, 8, 7, 4]).

# Acknowledgements

Ludwig Schmidt is supported by a Google PhD fellowship. Arturs Backurs is supported by an IBM Research fellowship. This research was supported by grants from NSF and Simons Foundation.

## Footnotes

[1]More efficient algorithms exist if the running time is allowed to be polynomial in the accuracy parameter, e.g., [35] give such an algorithm for the kernel SVM problem that we consider as well. See also the discussion at the end of this section.

[2]Consider a network with one hidden layer containing $n$ units and a training set with $m$ examples, for simplicity in small dimension $d = O(\log n)$. No known results preclude an algorithm that computes a full gradient in time $O((n+m)\log n)$. This would be significantly faster than the standard $O(n \cdot m \cdot \log n)$ approach of computing the full gradient example by example.

[3]Note that SETH can be viewed as a significant strengthening of the P $\neq$ NP conjecture, which only postulates that there is no *polynomial time* algorithm for CNF satisfiability. The best known algorithms for CNF satisfiability have running times of the form $O(2^{(1-o(1))n} \cdot \text{poly}(m))$.

[4]We use $\omega(g(n))$ to denote any function $f$ such that $\lim_{n \to \infty} f(n)/g(n) = \infty$. Similarly, we use $o(g(n))$ to denote any function $f$ such that $\lim_{n \to \infty} f(n)/g(n) = 0$. Consequently, we will refer to functions of the form $\omega(1)$ as *super-constant* and to $n^{\omega(1)}$ as *super-polynomial*.

[5]In the binary setting we consider, the logistic loss is equivalent to the softmax loss commonly employed in deep learning.

[6]Note that our reduction does not explicitly construct $M$. Instead, the values of the matrix are induced by the input examples and weights.

[7]We write $x \approx y$ if $x = y$ up to an inversely superpolynomial additive factor, i.e., $|x - y| \leq n^{-\omega(1)}$.

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
