[Supplementary Material]

# A  Preliminaries

In this section we define several notions used later in the paper. We start from the soft-margin support vector machine (see [25]).

**Definition 16** (Support Vector Machine (SVM))**.** *Let* $x_1, \ldots, x_n \in \mathbb{R}^d$ *be* $n$ *vectors and* $y_1, \ldots, y_n \in \{-1, 1\}$ *be* $n$ *labels. Let* $k(x, x')$ *be a kernel function. An optimization problem of the following form is a (primal) SVM.*

$$
\begin{aligned}
\underset{\substack{\alpha_1, \ldots, \alpha_n \geq 0, \ b \\ \xi_1, \ldots, \xi_n \geq 0}}{\text{minimize}} \quad & \frac{\lambda}{2} \sum_{i,j=1}^{n} \alpha_i \alpha_j y_i y_j k(x_i, x_j) \ + \ \frac{1}{n} \sum_{i=1}^{n} \xi_i \\
\text{subject to} \quad & y_i f(x_i) \geq 1 - \xi_i, \ \ i = 1, \ldots, n,
\end{aligned}
\tag{7}
$$

*where* $f(x) := b + \sum_{i=1}^{n} \alpha_i y_i k(x_i, x)$ *and* $\lambda \geq 0$ *is called the regularization parameter.* $\xi_i$ *are known as the slack variables.*

*The dual SVM is defined as*

$$
\begin{aligned}
\underset{\alpha_1, \ldots, \alpha_n \geq 0}{\text{maximize}} \quad & \sum_{i=1}^{n} \alpha_i - \frac{1}{2} \sum_{i,j=1}^{n} \alpha_i \alpha_j y_i y_j k(x_i, x_j) \\
\text{subject to} \quad & \sum_{i=1}^{n} \alpha_i y_i = 0, \\
& \alpha_1, \ldots, \alpha_n \leq \frac{1}{\lambda n}.
\end{aligned}
\tag{8}
$$

*We refer to the quantity* $b$ *as the bias term. When we require that the bias is* $b = 0$*, we call the optimization problem as SVM without the bias term. The primal SVM without the bias term remains the same except* $f(x) = \sum_{i=1}^{n} \alpha_i y_i k(x_i, x)$*. The dual SVM remains the same except we remove the equality constraint* $\sum_{i=1}^{n} \alpha_i y_i = 0$*.*

The (primal) hard-margin SVM defined in the previous section corresponds to soft-margin SVM in the setting when $\lambda \to 0$. The dual hard-margin SVM is defined as follows.

**Definition 17** (Dual hard-margin SVM)**.** *Let* $x_1, \ldots, x_n \in \mathbb{R}^d$ *be* $n$ *vectors and* $y_1, \ldots, y_n \in \{-1, 1\}$ *be* $n$ *labels. Let* $k(x, x')$ *be a kernel function. An optimization problem of the following form is a dual hard-margin SVM.*

$$
\begin{aligned}
\underset{\alpha_1, \ldots, \alpha_n \geq 0}{\text{maximize}} \quad & \sum_{i=1}^{n} \alpha_i - \frac{1}{2} \sum_{i,j=1}^{n} \alpha_i \alpha_j y_i y_j k(x_i, x_j) \\
\text{subject to} \quad & \sum_{i=1}^{n} \alpha_i y_i = 0.
\end{aligned}
\tag{9}
$$

*If the primal hard-margin SVM is without the bias term (* $b = 0$ *), then we omit the inequality constraint* $\sum_{i=1}^{n} \alpha_i y_i = 0$ *in the dual SVM.*

We will use the following fact (see [25]).

**Fact 18.** *If* $\alpha_1^*, \ldots, \alpha_n^*$ *achieve the minimum in an SVM, then the same* $\alpha_1^*, \ldots, \alpha_n^*$ *achieve the maximum in the dual SVM. Also, the minimum value and the maximum value are equal.*

# B  Hardness for SVM without the bias term

In this section we formalize the intuition from Section 4. We start from the following two lemmas.

**Lemma 19** (NO case)**.** *If for all* $a_i \in A$ *and* $b_j \in B$ *we have Hamming*$(a_i, b_j) \geq t$*, then*

$$
value(A, B) \leq value(A) + value(B) + 200n^6 \exp(-100 \log n \cdot t).
$$

**Lemma 20** (YES case). *If there exist $a_i \in A$ and $b_j \in B$ such that $Hamming(a_i, b_j) \leq t - 1$, then*

$$value(A, B) \geq value(A) + value(B) + \frac{1}{4}\exp(-100\log n \cdot (t-1)).$$

Assuming the two lemmas we can distinguish the NO case from the YES case because

$$200n^6 \exp(-100\log n \cdot t) \ll \frac{1}{4}\exp(-100\log n \cdot (t-1))$$

by our choice of the parameter $C = 100\log n$ for the Gaussian kernel.

Before we proceed with the proofs of the two lemmas, we prove the following auxiliary statement.

**Lemma 21.** *Consider SVM (3). Let $\alpha_1^*, \ldots, \alpha_n^*$ be the setting of values for $\alpha_1, \ldots, \alpha_n$ that achieves $value(A)$. Then for all $i = 1, \ldots, n$ we have that $n \geq \alpha_i^* \geq 1/2$.*

*Analogous statement holds for SVM (4).*

*Proof.* First we note that $value(A) \leq n^2/2$ because the objective value of (3) is at most $n^2/2$ if we set $\alpha_1 = \ldots = \alpha_n = 1$. Note that all inequalities of (3) are satisfied for this setting of variables. Now we lower bound $value(A)$:

$$value(A) = \frac{1}{2}\sum_{i,j}^n \alpha_i^* \alpha_j^* k(a_i, a_j) \geq \frac{1}{2}\sum_{i=1}^n (\alpha_i^*)^2.$$

From $value(A) \geq \frac{1}{2}\sum_{i=1}^n (\alpha_i^*)^2$ and $value(A) \leq n^2/2$ we conclude that $\alpha_i^* \leq n$ for all $i$.

Now we will show that $\alpha_i^* \geq 1/2$ for all $i = 1, \ldots, n$. Consider the inequality

$$\sum_{j=1}^n \alpha_j^* k(a_i, a_j) = \alpha_i^* + \sum_{j\,:\,j\neq i} \alpha_j^* k(a_i, a_j) \geq 1$$

which is satisfied by $\alpha_1^*, \ldots, \alpha_n^*$ because this is an inequality constraint in (3). Note that $k(a_i, a_j) \leq \frac{1}{10n^2}$ for all $i \neq j$ because $C = 100\log n$ and $\|a_i - a_j\|_2^2 = Hamming(a_i, a_j) \geq 1$ for all $i \neq j$. Also, we already obtained that $\alpha_j^* \leq n$ for all $j$. This gives us the required lower bound for $\alpha_i^*$:

$$\alpha_i^* \geq 1 - \sum_{j\,:\,j\neq i} \alpha_j^* k(a_i, a_j) \geq 1 - n \cdot n \cdot \frac{1}{10n^2} \geq 1/2.$$

$\square$

**Additive precision**  For particular value of $t$, the sufficient additive precision for solving the three SVMs is $\frac{1}{100}\exp(-100\log n \cdot (t-1))$ to be able to distinguish the NO case from the YES case. Since we want to be able to distinguish the two cases for any $t \in \{2, \ldots, d\}$, it suffices to have an additive precision $\exp(-100\log n \cdot d) \leq \frac{1}{100}\exp(-100\log n \cdot (t-1))$. From [5] we know that any $d = \omega(\log n)$ is sufficient to show hardness. Therefore, any additive approximation $\exp(-\omega(\log^2 n))$ is sufficient to show the hardness for SVM.

**Multiplicative precision**  Consider any $\varepsilon = \exp(-\omega(\log^2 n))$ and suppose we can approximate within multiplicative factor $(1 + \varepsilon)$ quantities $value(A)$, $value(B)$ and $value(A, B)$. From the proof of Lemma 21 we know that $value(A), value(B) \leq n^2/2$. If $value(A, B) \leq 10n^2$, then $(1 + \varepsilon)$-approximation of the three quantities allows us to compute the three quantities within additive $\exp(-\omega(\log^2 n))$ factor and the hardness follows from the previous paragraph. On the other hand, if $value(A, B) > 10n^2$, then $(1 + \varepsilon)$-approximation of $value(A, B)$ allows us to determine that we are in the YES case.

In the rest of the section we complete the proof of the theorem by proving Lemma 19 and Lemma 20.

*Proof of Lemma 19.* Let $\alpha_1^*, \ldots, \alpha_n^*$ and $\beta_1^*, \ldots, \beta_n^*$ be the optimal assignments to SVMs (3) and (4), respectively. We use the notation $\delta := \exp(-100\log n \cdot t)$. Note that $k(a_i, b_j) = \exp(-100\log n \cdot \|a_i - b_j\|_2^2) \leq \delta$ for all $i, j$ because $\|a_i - b_j\|_2^2 = Hamming(a_i, b_j) \geq t$ for all $i, j$.

We define $\alpha_i' := \alpha_i^* + 10n^2\delta$ and $\beta_i' := \beta_i^* + 10n^2\delta$ for all $i = 1, \ldots, n$. We observe that $\alpha_i', \beta_i' \leq 2n$ for all $i$ because $\alpha_i^*, \beta_i^* \leq n$ for all $i$ (Lemma 21) and $\delta = \exp(-100\log n \cdot t) \leq \frac{1}{10n^2}$. Let $V$ be the value of the objective function in (5) when evaluated on $\alpha_i'$ and $\beta_i'$.

We make two claims. We claim that $\alpha_i'$ and $\beta_i'$ satisfy the inequality constraints in (5). This implies that $\text{value}(A, B) \leq V$ since (5) is a minimization problem. Our second claim is that $V \leq \text{value}(A) + \text{value}(B) + 200n^6\delta$. The two claims combined complete the proof of the lemma.

We start with the proof of the second claim. We want to show that $V \leq \text{value}(A) + \text{value}(B) + 200n^6\delta$. We get the following inequality:

$$V = \frac{1}{2}\sum_{i,j=1}^{n}\alpha_i'\alpha_j'k(a_i, a_j) \ + \ \frac{1}{2}\sum_{i,j=1}^{n}\beta_i'\beta_j'k(b_i, b_j) \ - \ \sum_{i,j=1}^{n}\alpha_i'\beta_j'k(a_i, b_j)$$

$$\leq \frac{1}{2}\sum_{i,j=1}^{n}\alpha_i'\alpha_j'k(a_i, a_j) \ + \ \frac{1}{2}\sum_{i,j=1}^{n}\beta_i'\beta_j'k(b_i, b_j)$$

since the third sum is non-negative. It is sufficient to show two inequalities $\frac{1}{2}\sum_{i,j=1}^{n}\alpha_i'\alpha_j'k(a_i, a_j) \leq \text{value}(A) + 100n^6\delta$ and $\frac{1}{2}\sum_{i,j=1}^{n}\beta_i'\beta_j'k(b_i, b_j) \leq \text{value}(B) + 100n^6\delta$ to establish the inequality $V \leq \text{value}(A) + \text{value}(B) + 200n^6\delta$. We prove the first inequality. The proof for the second inequality is analogous. We use the definition of $\alpha_i' = \alpha_i^* + 10n^2\delta$:

$$\frac{1}{2}\sum_{i,j=1}^{n}\alpha_i'\alpha_j'k(a_i, a_j)$$

$$= \frac{1}{2}\sum_{i,j=1}^{n}(\alpha_i^* + 10n^2\delta)(\alpha_j^* + 10n^2\delta)k(a_i, a_j)$$

$$\leq \frac{1}{2}\sum_{i,j=1}^{n}\left(\alpha_i^*\alpha_j^*k(a_i, a_j) + 20n^3\delta + 100n^4\delta^2\right)$$

$$\leq \text{value}(A) + 100n^6\delta,$$

where in the first inequality we use that $\alpha_i^* \leq n$ and $k(a_i, a_j) \leq 1$.

Now we prove the first claim. We show that the inequality constraints are satisfied by $\alpha_i'$ and $\beta_i'$. We prove that the inequality

$$\sum_{j=1}^{n}\alpha_j'k(a_i, a_j) - \sum_{j=1}^{n}\beta_j'k(a_i, b_j) \geq 1 \tag{10}$$

is satisfied for all $i = 1, \ldots, n$. The proof that the inequalities $-\sum_{j=1}^{n}\beta_j'k(b_i, b_j) + \sum_{j=1}^{n}\alpha_j'k(b_i, a_j) \leq -1$ are satisfied is analogous.

We lower bound the first sum of the left hand side of (10) by repeatedly using the definition of $\alpha_i' = \alpha_i^* + 10n^2\delta$:

$$\sum_{j=1}^{n}\alpha_j'k(a_i, a_j)$$

$$= (\alpha_i^* + 10n^2\delta) + \sum_{j\,:\,j\neq i}\alpha_j'k(a_i, a_j)$$

$$\geq 10n^2\delta + \alpha_i^* + \sum_{j\,:\,j\neq i}\alpha_j^*k(a_i, a_j)$$

$$= 10n^2\delta + \sum_{j=1}^{n}\alpha_j^*k(a_i, a_j)$$

$$\geq 1 + 10n^2\delta.$$

In the last inequality we used the fact that $\alpha_i^*$ satisfy the inequality constraints of SVM (3).

We upper bound the second sum of the left hand side of (10) by using the inequality $\beta'_j \leq 2n$ and $k(a_i, b_j) \leq \delta$ for all $i, j$:

$$\sum_{j=1}^{n} \beta'_j k(a_i, b_j) \leq 2n^2 \delta.$$

Finally, we can show that the inequality constraint is satisfied:

$$\sum_{j=1}^{n} \alpha'_j k(a_i, a_j) - \sum_{j=1}^{n} \beta'_j k(a_i, b_j) \geq 1 + 10n^2\delta - 2n^2\delta \geq 1.$$

$\square$

*Proof of Lemma 20.* To analyze the YES case, we consider the dual SVMs (see Definition 17) of the three SVMs (3), (4) and (5):

1. The dual SVM of SVM (3):

$$\underset{\alpha_1,\ldots,\alpha_n \geq 0}{\text{maximize}} \quad \sum_{i=1}^{n} \alpha_i - \frac{1}{2} \sum_{i,j=1}^{n} \alpha_i \alpha_j k(a_i, a_j). \tag{11}$$

2. The dual SVM of SVM (4):

$$\underset{\beta_1,\ldots,\beta_n \geq 0}{\text{maximize}} \quad \sum_{i=1}^{n} \beta_i - \frac{1}{2} \sum_{i,j=1}^{n} \beta_i \beta_j k(a_i, a_j). \tag{12}$$

3. The dual SVM of SVM (5):

$$\underset{\substack{\alpha_1,\ldots,\alpha_n \geq 0 \\ \beta_1,\ldots,\beta_n \geq 0}}{\text{maximize}} \quad \sum_{i=1}^{n} \alpha_i + \sum_{i=1}^{n} \beta_i - \frac{1}{2} \sum_{i,j=1}^{n} \alpha_i \alpha_j k(a_i, a_j) - \frac{1}{2} \sum_{i,j=1}^{n} \beta_i \beta_j k(b_i, b_j) + \sum_{i,j=1}^{n} \alpha_i \beta_j k(a_i, b_j). \tag{13}$$

Since the optimal values of the primal and dual SVMs are equal, we have that value$(A)$, value$(B)$ and value$(A, B)$ are equal to optimal values of dual SVMs (11), (12) and (13), respectively (see Fact 18).

Let $\alpha_1^*, \ldots, \alpha_n^*$ and $\beta_1^*, \ldots, \beta_n^*$ be the optimal assignments to dual SVMs (11) and (12), respectively.

Our goal is to lower bound value$(A, B)$. Since (13) is a maximization problem, it is sufficient to show an assignment to $\alpha_i$ and $\beta_j$ that gives a large value to the objective function. For this we set $\alpha_i = \alpha_i^*$ and $\beta_j = \beta_j^*$ for all $i, j = 1, \ldots, n$. This gives the following inequality:

$$\text{value}(A, B) \geq \sum_{i=1}^{n} \alpha_i^* + \sum_{i=1}^{n} \beta_i^* - \frac{1}{2} \sum_{i,j=1}^{n} \alpha_i^* \alpha_j^* k(a_i, a_j) - \frac{1}{2} \sum_{i,j=1}^{n} \beta_i^* \beta_j^* k(b_i, b_j) + \sum_{i,j=1}^{n} \alpha_i^* \beta_j^* k(a_i, b_j)$$

$$\geq \text{value}(A) + \text{value}(B) + \sum_{i,j=1}^{n} \alpha_i^* \beta_j^* k(a_i, b_j),$$

where we use the fact that value$(A)$ and value$(B)$ are the optimal values of dual SVMs (11) and (12), respectively.

To complete the proof of the lemma, it suffices to show the following inequality:

$$\sum_{i,j=1}^{n} \alpha_i^* \beta_j^* k(a_i, b_j) \geq \frac{1}{4} \exp(-100 \log n \cdot (t-1)). \tag{14}$$

Notice that so far we did not use the fact that there is a close pair of vectors $a_i \in A$ and $b_j \in B$ such that Hamming$(a_i, b_j) \leq t - 1$. We use this fact now. We lower bound the left hand side of (14) by the summand corresponding to the close pair:

$$\sum_{i,j=1}^{n} \alpha_i^* \beta_j^* k(a_i, b_j) \geq \alpha_i^* \beta_j^* k(a_i, b_j) \geq \alpha_i^* \beta_j^* \exp(-100 \log n \cdot (t-1)),$$

where in the last inequality we use $\text{Hamming}(a_i, b_j) \leq t - 1$ and the definition of the Gaussian kernel.

The proof is completed by observing that $\alpha_i^* \geq \frac{1}{2}$ and $\beta_i^* \geq \frac{1}{2}$ which follows from Fact 18 and Lemma 21. $\qquad\square$

## C  Hardness for SVM with the bias term

In the previous section we showed hardness for SVM without the bias term. In this section we show hardness for SVM with the bias term.

**Theorem 22.** *Let $x_1, \ldots, x_n \in \{-1, 0, 1\}^d$ be $n$ vectors and let $y_1, \ldots, y_n \in \{-1, 1\}$ be $n$ labels.*

*Let $k(a, a') = \exp\left(-C\|a - a'\|_2^2\right)$ be the Gaussian kernel with $C = 100 \log n$.*

*Consider the corresponding hard-margin SVM with the bias term:*

$$
\begin{aligned}
\underset{\alpha_1, \ldots, \alpha_n \geq 0, \ b}{\text{minimize}} \quad & \frac{1}{2} \sum_{i,j=1}^{n} \alpha_i \alpha_j y_i y_j k(x_i, x_j) \\
\text{subject to} \quad & y_i f(x_i) \geq 1, \quad i = 1, \ldots, n,
\end{aligned}
\tag{15}
$$

*where $f(x) := b + \sum_{i=1}^{n} \alpha_i y_i k(x_i, x)$.*

*Consider any $\varepsilon = \exp(-\omega(\log^2 n))$. Approximating the optimal value of* (15) *within the multiplicative factor $(1 + \varepsilon)$ requires almost quadratic time assuming SETH. This holds for the dimensionality $d = O(\log^3 n)$ of the input vectors.*

*The same hardness result holds for any additive $\exp(-\omega(\log^2 n))$ approximation factor.*

*Proof.* Consider a hard instance from Theorem 4 for SVM without the bias term. Let $x_1, \ldots, x_n \in \{0, 1\}^d$ be the $n$ binary vectors of dimensionality $d = \omega(\log n)$ and $y_1, \ldots, y_n \in \{-1, 1\}$ be the $n$ corresponding labels. For this input consider the dual SVM without the bias term (see Definition 17):

$$
\underset{\gamma_1, \ldots, \gamma_n \geq 0}{\text{maximize}} \quad \sum_{i=1}^{n} \gamma_i - \frac{1}{2} \sum_{i,j=1}^{n} \gamma_i \gamma_j y_i y_j k(x_i, x_j).
\tag{16}
$$

We will show how to reduce SVM without the bias term (16) to SVM with the bias term. By Theorem 4 this will give hardness result for SVM with the bias term. We start with a simpler reduction that will achieve almost what we need except the entries of the vectors will not be from the set $\{-1, 0, 1\}$. Then we will show how to change the reduction to fix this.

Consider $2n$ vectors $x_1, \ldots, x_n, -x_1, \ldots, -x_n \in \{-1, 0, 1\}^d$ with $2n$ labels $y_1, \ldots, y_n, -y_1, \ldots, -y_n \in \{-1, 1\}$. Consider an SVM with the bias term for the $2n$ vectors, that is, an SVM of the form (15). From Definition 17 we know that its dual SVM is

$$
\begin{aligned}
\underset{\substack{\alpha_1, \ldots, \alpha_n \geq 0 \\ \beta_1, \ldots, \beta_n \geq 0}}{\text{maximize}} \quad & \sum_{i=1}^{n} \alpha_i + \sum_{j=1}^{n} \beta_j \\
& - \frac{1}{2} \sum_{i,j=1}^{n} \alpha_i \alpha_j y_i y_j k(x_i, x_j) - \frac{1}{2} \sum_{i,j=1}^{n} \beta_i \beta_j y_i y_j k(x_i, x_j) + \sum_{i,j=1}^{n} \alpha_i \beta_j y_i y_j k(x_i, -x_j) \\
\text{subject to} \quad & \sum_{i=1}^{n} \alpha_i y_i = \sum_{j=1}^{n} \beta_j y_j.
\end{aligned}
\tag{17}
$$

Consider any setting of values for $\alpha_i$ and $\beta_j$. Notice that if we swap the value of $\alpha_i$ and $\beta_i$ for every $i$, the value of the objective function of (17) does not change. This is implies that we can define $\gamma_i := \frac{\alpha_i + \beta_i}{2}$ and set $\alpha_i = \beta_i = \gamma_i$ for every $i$. Because of the convexity of the optimization problem, the value of the objective function can only increase after this change. Clearly, the equality constraint

will be satisfied. Therefore, w.l.o.g. we can assume that $\alpha_i = \beta_i = \gamma_i$ for some $\gamma_i$ and we can omit the equality constraint.

We rewrite (17) in terms of $\gamma_i$ and divide the objective function by 2:

$$\underset{\gamma_1,\ldots,\gamma_n \geq 0}{\text{maximize}} \quad \sum_{i=1}^n \gamma_i \; - \; \frac{1}{2} \sum_{i,j=1}^n \gamma_i \gamma_j y_i y_j k(x_i, x_j) \; + \; \frac{1}{2} \sum_{i,j=1}^n \gamma_i \gamma_j y_i y_j k(x_i, -x_j). \tag{18}$$

Notice that (18) and (16) are almost the same. The only difference is the third term

$$\frac{1}{2} \sum_{i,j=1}^n \gamma_i \gamma_j y_i y_j k(x_i, -x_j)$$

in (18). We can make this term to be equal to 0 and not change the first two terms as follows. We append an extra coordinate to every vector $x_i$ and set this coordinate to be large enough value $M$. If we set $M = +\infty$, the third term becomes 0. The first term does not depend on the vectors. The second term depends only on the distances between the vectors (which are not affected by adding the same entry to all vectors). Thus, the first two terms do not change after this modification.

We showed that we can reduce SVM without the bias term (16) to the SVM with the bias term (17). By combining this reduction with Theorem 4 we obtain hardness for SVM with the bias term. This is almost what we need except that the reduction presented above produces vectors with entries that are not from the set $\{-1, 0, 1\}$. In every vector $x_i$ or $-x_i$ there is an entry that has value $M$ or $-M$, respectively. In the rest of the proof we show how to fix this, by bounding $M$ by $O(\log^3 n)$ and distributing its contribution over $O(\log^3 n)$ coordinates.

**Final reduction** The final reduction is as follows:

- Take a hard instance for the SVM without the bias term from Theorem 4. Let $x_1, \ldots, x_n \in \{0, 1\}^d$ be the $n$ binary vectors of dimensionality $d = \omega(\log n)$ and $y_1, \ldots, y_n \in \{-1, 1\}$ be the $n$ corresponding labels.

- Append $\log^3 n$ entries to each of the vectors $x_i$, $i = 1, \ldots, n$ and set the entries to be 1.

- Solve SVM (15) on the $2n$ vectors $x_1, \ldots, x_n, -x_1, \ldots, -x_n \in \{-1, 0, 1\}^d$ with $2n$ labels $y_1, \ldots, y_n, -y_1, \ldots, -y_n \in \{-1, 1\}$. Let $V$ be the optimal value of the SVM divided by 2.

- Output $V$.

**Correctness of the reduction** From the above discussion we know that we output the optimal value $V$ of the optimization problem (18). Let $V'$ be the optimal value of SVM (16).

By Theorem 4, it is sufficient to show that $|V - V'| \leq \exp(-\omega(\log^2 n))$ to establish hardness for SVM with the bias term. We will show that $|V - V'| \leq n^{O(1)} \exp(-\log^3 n)$. This gives hardness for additive approximation of SVM with the bias term. However, $|V - V'| \leq \exp(-\omega(\log^2 n))$ is also sufficient to show hardness for multiplicative approximation (see the discussion on the approximation in the proof of Theorem 4).

In the rest of the section we show that $|V - V'| \leq n^{O(1)} \exp(-\log^3 n)$. Let $\gamma_i'$ be the assignment to $\gamma_i$ that achieves $V'$ in SVM (16). Let $\gamma_i^*$ be the assignment to $\gamma_i$ that achieves $V$ in (18). We will show that $\gamma_i' \leq O(n)$ for all $i = 1, \ldots, n$. It is also true that $\gamma_i^* \leq O(n)$ for all $i = 1, \ldots, n$ and the proof is analogous. Since $x_1, \ldots, x_n$ are different binary vectors and $k(x_i, x_j)$ is the Gaussian kernel with the parameter $C = 100 \log n$, we have that $k(x_i, x_j) \leq 1/n^{10}$ for all $i \neq j$. This gives the following upper bound:

$$V' = \sum_{i=1}^n \gamma_i' \; - \; \frac{1}{2} \sum_{i,j=1}^n \gamma_i' \gamma_j' y_i y_j k(x_i, x_j) \leq \sum_{i=1}^n \left( \gamma_i' - \left( \frac{1}{2} - o(1) \right) (\gamma_i')^2 \right).$$

Observe that every non-negative summand on the right hand side is at most $O(1)$. Therefore, if there exists $i$ such that $\gamma_i' \geq \omega(n)$, then the right hand side is negative. This contradicts the lower bound $V' \geq 0$ (which follows by setting all $\gamma_i$ to be 0 in (16)).

By plugging $\gamma_i'$ into (18) and using the fact that $\gamma_i' \leq O(n)$, we obtain the following inequality:

$$V \geq V' + \frac{1}{2} \sum_{i,j=1}^{n} \gamma_i' \gamma_j' y_i y_j k(x_i, -x_j) \geq V' - n^{O(1)} \exp(-\log^3 n). \tag{19}$$

In the last inequality we use $k(x_i, -x_j) \leq \exp(-\log^3 n)$ which holds for all $i, j = 1, ..., n$ (observe that each $x_i$ and $x_j$ ends with $\log^3 n$ entries 1 and use the definition of the Gaussian kernel).

Similarly, by plugging $\gamma_i^*$ into (16) and using the fact that $\gamma_i^* \leq O(n)$, we obtain the following inequality:

$$V' \geq V - \frac{1}{2} \sum_{i,j=1}^{n} \gamma_i^* \gamma_j^* y_i y_j k(x_i, -x_j) \geq V - n^{O(1)} \exp(-\log^3 n). \tag{20}$$

Inequalities (19) and (20) combined give the desired inequality $|V - V'| \leq n^{O(1)} \exp(-\log^3 n)$. $\quad\square$

## D   Hardness for soft-margin SVM

**Theorem 23.** *Let $x_1, \ldots, x_n \in \{-1, 0, 1\}^d$ be $n$ vectors and let $y_1, \ldots, y_n \in \{-1, 1\}$ be $n$ labels.*

*Let $k(a, a') = \exp\left(-C\|a - a'\|_2^2\right)$ be the Gaussian kernel with $C = 100 \log n$.*

*Consider the corresponding soft-margin SVM with the bias term:*

$$\begin{array}{ll} \underset{\substack{\alpha_1, \ldots, \alpha_n \geq 0, \ b \\ \xi_1, \ldots, \xi_n \geq 0}}{minimize} & \dfrac{\lambda}{2} \sum_{i,j=1}^{n} \alpha_i \alpha_j y_i y_j k(x_i, x_j) \ + \ \dfrac{1}{n} \sum_{i=1}^{n} \xi_i \\[2ex] subject\ to & y_i f(x_i) \geq 1 - \xi_i, \ \ i = 1, \ldots, n, \end{array} \tag{21}$$

*where $f(x) := b + \sum_{i=1}^{n} \alpha_i y_i k(x_i, x)$.*

*Consider any $\varepsilon = \exp(-\omega(\log^2 n))$ and any $0 < \lambda \leq \frac{1}{Kn^2}$ for a large enough constant $K > 0$. Approximating the optimal value of (21) within the multiplicative factor $(1 + \varepsilon)$ requires almost quadratic time assuming SETH. This holds for the dimensionality $d = O(\log^3 n)$ of the input vectors.*

*The same hardness result holds for any additive $\exp(-\omega(\log^2 n))$ approximation factor.*

*Proof.* Consider the hard instance from Theorem 22 for the hard-margin SVM. The dual of the hard-margin SVM is (17). From the proof we know that the optimal $\alpha_i$ and $\beta_i$ satisfy $\alpha_i = \beta_i = \gamma_i^* \leq 2Kn$ for some large enough constant $K > 0$ for all $i = 1, \ldots, n$. Thus, w.l.o.g. we can add these inequalities to the set of constraints. We compare the resulting dual SVM to Definition 16 and conclude that the resulting dual SVM is a dual of a *soft-margin* SVM with the regularization parameter $\lambda = \frac{1}{Kn^2}$. Therefore, the hardness follows from Theorem 22. $\quad\square$

## E   Hardness proof for Kernel PCA

In this section, we present the full proof of quadratic hardness for Kernel PCA. It will also be helpful for Kernel Ridge Regression in the next section.

Given a matrix $X$, we denote its trace (the sum of the diagonal entries) by $\operatorname{tr}(X)$ and the total sum of its entries by $s(X)$. In the context of the matrix $K'$ defining our problem, we have the following equality:

$$\begin{aligned} \operatorname{tr}(K') &= \operatorname{tr}((I - 1_n)K(I - 1_n)) \\ &= \operatorname{tr}(K(I - 1_n)^2) = \operatorname{tr}(K(I - 1_n)) \\ &= \operatorname{tr}(K) - \operatorname{tr}(K1_n) = n - s(K)/n \, . \end{aligned}$$

Since the sum of the eigenvalues is equal to the trace of the matrix and $\operatorname{tr}(K') = n - s(K)/n$, it is sufficient to show hardness for computing $s(K)$. The following lemma completes the proof of the theorem.

**Lemma 24.** *Computing $s(K)$ within multiplicative error $1 + \varepsilon$ for $\varepsilon = \exp(-\omega(\log^2 n))$ requires almost quadratic time assuming SETH.*

*Proof.* As for SVMs, we will reduce the BHCP problem to the computation of $s(K)$. Let $A$ and $B$ be the two sets of $n$ binary vectors coming from an instance of the BHCP problem. Let $K_A, K_B \in \mathbb{R}^{n \times n}$ be the kernel matrices corresponding to the sets $A$ and $B$, respectively. Let $K_{A,B} \in \mathbb{R}^{2n \times 2n}$ be the kernel matrix corresponding to the set $A \cup B$. We observe that

$$s := (s(K_{A,B}) - s(K_A) - s(K_B))/2$$
$$= \sum_{i,j=1}^{n} k(a_i, b_j)$$
$$= \sum_{i,j=1}^{n} \exp(-C||a_i - b_j||_2^2).$$

Now we consider two cases.

**Case 1.** There are no close pairs, that is, for all $i, j = 1, \ldots, n$ we have $||a_i - b_j||_2^2 \geq t$ and $\exp(-C||a_i - b_j||_2^2) \leq \exp(-Ct) =: \delta$. Then $s \leq n^2 \delta$.

**Case 2.** There is a close pair. That is, $||a_{i'} - b_{j'}||_2^2 \leq t - 1$ for some $i', j'$. This implies that $\exp(-C||a_{i'} - b_{j'}||_2^2) \geq \exp(-C(t-1)) =: \Delta$. Thus, $s \geq \Delta$.

Since $C = 100 \log n$, we have that $\Delta \geq n^{10} \delta$ and we can distinguish the two cases.

**Precision.** To distinguish $s \geq \Delta$ from $s \leq n^2 \delta$, it is sufficient that $\Delta \geq 2n^2 \delta$. This holds for $C = 100 \log n$. The sufficient additive precision is $\exp(-Cd) = \exp(-\omega(\log^2 n))$. Since $s(K) \leq O(n^2)$ for any Gaussian kernel matrix $K$, we also get that $(1 + \varepsilon)$ multiplicative approximation is sufficient to distinguish the cases for any $\varepsilon = \exp(-\omega(\log^2 n))$. $\square$

# F    Hardness for kernel ridge regression

We start with stating helpful definitions and lemmas.

We will use the following lemma which is a consequence of the binomial inverse theorem.

**Lemma 25.** *Let $X$ and $Y$ be two square matrices of equal size. Then the following equality holds:*

$$(X + Y)^{-1} = X^{-1} - X^{-1}(I + YX^{-1})^{-1}YX^{-1}.$$

**Definition 26** (Almost identity matrix). *Let $X \in \mathbb{R}^{n \times n}$ be a matrix. We call it* almost identity matrix *if $X = I + Y$ and $|Y_{i,j}| \leq n^{-\omega(1)}$ for all $i, j = 1, \ldots, n$.*

We will need the following two lemmas.

**Lemma 27.** *The product of two almost identity matrices is an almost identity matrix.*

*Proof.* Follows easily from the definition. $\square$

**Lemma 28.** *The inverse of an almost identity matrix is an almost identity matrix.*

*Proof.* Let $X$ be an almost identity matrix. We want to show that $X^{-1}$ is an almost identity matrix. We write $X = I - Y$ such that $|Y_{i,j}| \leq n^{-\omega(1)}$ for all $i, j = 1, \ldots, n$. We have the following matrix equality

$$X^{-1} = (I - Y)^{-1} = I + Y + Y^2 + Y^3 + \ldots$$

To show that $X^{-1}$ is an almost identity, we will show that the absolute value of every entry of $Z := Y + Y^2 + Y^3 + \ldots$ is at most $n^{-\omega(1)}$. Let $\varepsilon \leq n^{-\omega(1)}$ is an upper bound on $|Y_{i,j}|$ for all $i, j = 1, \ldots, n$. Then $|Z_{i,j}| \leq Z'_{i,j}$, where $Z' := Y' + (Y')^2 + (Y')^3 + \ldots$ and $Y'$ is a matrix consisting of entries that are all equal to $\varepsilon$. The proof follows since $Z'_{i,j} = \sum_{k=1}^{\infty} \varepsilon^k n^{k-1} \leq 10\varepsilon \leq n^{-\omega(1)}$. $\square$

In the rest of the section we prove Theorem 6.

*Proof of Theorem 6.* We reduce the BHCP problem to the problem of computing the sum of the entries of $K^{-1}$.

Let $A$ and $B$ be the two sets of binary vectors from the BHCP instance. Let $K \in \mathbb{R}^{2n \times 2n}$ be the corresponding kernel matrix. We can write the kernel matrix $K$ as combination of four smaller matrices $K^{1,1}, K^{1,2}, K^{2,1}, K^{2,2} \in \mathbb{R}^{n \times n}$:

$$K = \left[ \begin{array}{c|c} K^{1,1} & K^{1,2} \\ \hline K^{2,1} & K^{2,2} \end{array} \right].$$

$K^{1,1}$ is the kernel matrix for the set of vectors $A$ and $K^{2,2}$ is the kernel matrix for the set of vectors $B$. We define two new matrices $X, Y \in \mathbb{R}^{2n \times 2n}$: $X = \left[ \begin{array}{c|c} K^{1,1} & 0 \\ \hline 0 & K^{2,2} \end{array} \right]$ and $Y = \left[ \begin{array}{c|c} 0 & K^{1,2} \\ \hline K^{2,1} & 0 \end{array} \right]$.

For any matrix $Z$, let $s(Z)$ denote the sum of all entries of $Z$. Using Lemma 25, we can write $K^{-1}$ as follows:

$$K^{-1} = (X + Y)^{-1} = X^{-1} - X^{-1}(I + YX^{-1})^{-1}YX^{-1}.$$

We note that the matrix $X$ is an almost identity and that $|Y_{i,j}| \le n^{-\omega(1)}$ for all $i, j = 1, \ldots, 2n$. This follows from the fact that we use the Gaussian kernel function with the parameter $C = \omega(\log n)$ and the input vectors are binary. Combining this with lemmas 27 and 28 allows us to conclude that matrices $X^{-1}(I + YX^{-1})^{-1}$ and $X^{-1}$ are almost identity. Since all entries of the matrix $Y$ are non-negative, we conclude that

$$s(X^{-1}(I + YX^{-1})^{-1}YX^{-1}) = s(Y)(1 \pm n^{-\omega(1)}).$$

We obtain that

$$\begin{aligned} s(K^{-1}) &= s(X^{-1}) - s(X^{-1}(I + YX^{-1})^{-1}YX^{-1}) \\ &= s(X^{-1}) - s(Y)(1 \pm n^{-\omega(1)}) \\ &= s\left((K^{1,1})^{-1}\right) + s\left((K^{2,2})^{-1}\right) - s(Y)(1 \pm n^{-\omega(1)}). \end{aligned}$$

Fix any $\alpha = \exp(-\omega(\log^2 n))$. Suppose that we can estimate each $s(K^{-1})$, $s\left((K^{1,1})^{-1}\right)$ and $s\left((K^{2,2})^{-1}\right)$ within the additive factor of $\alpha$. This allows us to estimate $s(Y)$ within the additive factor of $10\alpha$. This is enough to solve the BHCP problem. We consider two cases.

**Case 1** There are no close pairs, that is, for all $i, j = 1, \ldots, n$ we have $\|a_i - b_j\|_2^2 \ge t$ and $\exp(-C\|a_i - b_j\|_2^2) \le \exp(-Ct) =: \delta$. Then $s(Y) \le 2n^2\delta$.

**Case 2** There is a close pair. That is, $\|a_{i'} - b_{j'}\|_2^2 \le t - 1$ for some $i', j'$. This implies that $\exp(-C\|a_{i'} - b_{j'}\|_2^2) \ge \exp(-C(t-1)) =: \Delta$. Thus, $s(Y) \ge \Delta$.

Since $C = \omega(\log n)$, we have that $\Delta \ge 100n^2\delta$ and we can distinguish the two cases assuming that the additive precision $\alpha = \exp(-\omega(\log^2 n))$ is small enough.

**Precision** To distinguish $s(Y) \le 2n^2\delta$ from $s(Y) \ge \Delta$, it is sufficient that $\Delta \ge 100n^2\delta$ and $\alpha \le \Delta/1000$. We know that $\Delta \ge 100n^2\delta$ holds because $C = \omega(\log n)$. Since $\Delta \le \exp(-Cd)$, we want to choose $C$ and $d$ such that the $\alpha \le \Delta/1000$ is satisfied. We can do that because we can pick $C$ to be any $C = \omega(\log n)$ and the BHCP problem requires almost quadratic time assuming SETH for any $d = \omega(\log n)$.

We get that additive $\varepsilon$ approximation is sufficient to distinguish the cases for any $\varepsilon = \exp(-\omega(\log^2 n))$. We observe that $s(K^{-1}) \le O(n)$ for any almost identity matrix $K$. This means that $(1 + \varepsilon)$ multiplicative approximation is sufficient for the same $\varepsilon$. This completes the proof of the theorem. □

# G Hardness for training of the final layer of a neural network

Recall that the trainable parameters are $\alpha := (\alpha_1, \ldots, \alpha_n)^\mathsf{T}$, and that the optimization problem (2) is equivalent to the following optimization problem:

$$\underset{\alpha \in \mathbb{R}^n}{\text{minimize}} \quad \sum_{i=1}^{m} l(y_i \cdot (M\alpha)_i), \tag{22}$$

where $M \in \mathbb{R}^{m \times n}$ is the matrix defined as $M_{i,j} := S(u_i^\mathsf{T} w_j)$ for $i = 1, \ldots, m$ and $j = 1, \ldots n$. For the rest of the section we will use $m = \Theta(n)$.

Let $A = \{a_1, \ldots, a_n\} \subseteq \{0,1\}^d$ and $B = \{b_1, \ldots, b_n\} \subseteq \{0,1\}^d$ with $d = \omega(\log n)$ be the input to the Orthogonal Vectors problem. We construct a matrix $M$ as a combination of three smaller matrices:

$$M = \begin{bmatrix} M_1 \\ M_2 \\ M_2 \end{bmatrix}.$$

Both matrices $M_1, M_2 \in \mathbb{R}^{n \times n}$ are of size $n \times n$. Thus we have that the number of rows of $M$ is $m = 3n$.

We describe the two matrices $M_1, M_2$ below. Recall that $v_0, v_1$, and $v_2$ are given in Definition 12.

- $(M_1)_{i,j} = S\left(v_0 - (v_2 - v_0) \cdot a_i^\mathsf{T} b_j\right)$. For any two real values $x, y \in \mathbb{R}$ we write $x \approx y$ if $x = y$ up to an inversely superpolynomial additive factor. In other words, $|x - y| \leq n^{-\omega(1)}$. We observe that if two vectors $a_i$ and $b_j$ are orthogonal, then the corresponding entry $(M_1)_{i,j} = S(v_0) = \Theta(1)$ and otherwise $(M_1)_{i,j} \approx 0$. We will show that an $\left(1 + \frac{1}{4n}\right)$-approximation of the optimal value of the optimization problem (22) will allow us to decide whether there is an entry in $M_1$ that is $S(v_0) = \Theta(1)$. This will give the required hardness. It remains to show how to construct the matrix $M_1$ using a neural network. We set the weights for the $j$-th hidden unit to be $\begin{bmatrix} b_j \\ 1 \end{bmatrix}$. That is, $d$ weights are specified by the vector $b_j$, and we add one more input with weight 1. The $i$-th example (corresponding to the $i$-th row of the matrix $M_1$) is the vector $\begin{bmatrix} -(v_2 - v_0)a_i \\ v_0 \end{bmatrix}$. The output of the $j$-th unit on this example (which corresponds to entry $(M_1)_{i,j}$) is equal to

$$S\left(\begin{bmatrix} -(v_2 - v_0)a_i \\ v_0 \end{bmatrix}^\mathsf{T} \begin{bmatrix} b_j \\ 1 \end{bmatrix}\right) = S\left(v_0 - (v_2 - v_0) \cdot a_i^\mathsf{T} b_j\right)$$
$$= (M_1)_{i,j}$$

as required.

- $(M_2)_{i,j} = S\left(v_1 - (v_2 - v_1) \cdot \bar{b}_i^\mathsf{T} b_j\right)$, where $\bar{b}_i$ is a binary vector obtained from the binary vector $b_i$ by complementing all bits. We observe that this forces the diagonal entries of $M_2$ to be equal to $(M_2)_{i,i} = S(v_1) = 1/n^{1000K}$ for all $i = 1, \ldots, n$ and the off-diagonal entries to be $(M_2)_{i,j} \approx 0$ for all $i \neq j$.[9]

To complete the description of the optimization problem (22), we assign labels to the inputs corresponding to the rows of the matrix $M$. We assign label 1 to all inputs corresponding to rows of the matrix $M_1$ and the first copy of the matrix $M_2$. We assign label $-1$ to all remaining rows of the matrix $M$ corresponding to the second copy of matrix $M_2$.

It now suffices to prove Lemma 13 and Lemma 14.

*Proof of Lemma 13.* To obtain an upper bound on the optimal value in the presence of an orthogonal pair, we set the vector $\alpha$ to have all entries equal to $n^{100K}$. For this $\alpha$ we have

- $|(M_1\alpha)_i| \geq \Omega(n^{100K})$ for all $i = 1, \ldots, n$ such that there is exists $j = 1, \ldots, n$ with $a_i^\mathsf{T} b_j = 0$. Let $x \geq 1$ be the number of such $i$.

- $|(M_1\alpha)_i| \leq n^{-\omega(1)}$ for all $i = 1, \ldots, n$ such that there is no $j = 1, \ldots, n$ with $a_i^\mathsf{T} b_j = 0$. The number of such $i$ is $n - x$.

By using the second property of Definition 11, the total loss corresponding to $M_1$ is upper bounded by

$$x \cdot l(\Omega(n^{100K})) + (n - x) \cdot l(n^{-\omega(1)}) \leq x \cdot o(1) + (n - x) \cdot (l(0) + o(1/n))$$
$$\leq (n - 1) \cdot l(0) + o(1) =: l_1.$$

Finally, the total loss corresponding to the two copies of the matrix $M_2$ is upper bounded by

$$2n \cdot l(\pm O(n^{-800K})) = 2n \cdot (l(0) \pm o(1/n))$$
$$\leq 2n \cdot l(0) + o(1) =: l_2.$$

The total loss corresponding to the matrix $M$ is upper bounded by $l_1 + l_2 \leq (3n - 1) \cdot l(0) + o(1)$ as required. □

*Proof of Lemma 14.* We first observe that the total loss corresponding to the two copies of the matrix $M_2$ is lower bounded by $2n \cdot l(0)$. Consider the $i$-th row in both copies of matrix $M_2$. By using the convexity of the function $l$, the loss corresponding to the two rows is lower bounded by $l((M_2\alpha)_i) + l(-(M_2\alpha)_i) \geq 2 \cdot l(0)$. By summing over all $n$ pairs of rows we obtain the required lower bound on the loss.

We claim that $\|\alpha\|_\infty \leq n^{10^6 K}$. Suppose that this is not the case and let $i$ be the index of the largest entry of $\alpha$ in magnitude. Then the $i$-th entry of the vector $M_2\alpha$ is

$$(M_2\alpha)_i = \alpha_i (M_2)_{i,i} \pm n \cdot \alpha_i \cdot n^{-\omega(1)}$$
$$\geq \frac{\alpha_i}{n^{1000K}} - \alpha_i n^{-\omega(1)},$$

where we recall that the diagonal entries of matrix $M_2$ are equal to $(M_2)_{i,i} = S(v_1) = 1/n^K$. If $|\alpha_i| > n^{10^6 K}$, then $|(M_2\alpha)_i| \geq n^{1000k}$. However, by the second property in Definition 11, this implies that the loss is lower bounded by $\omega(n)$ for the $i$-row (for the first or the second copy of $M_2$). This contradicts a simple lower bound of $4n \cdot l(0)$ on the loss obtained by setting $\alpha = 0$ to be the all 0s vector. We use the third property of a nice loss function which says that $l(0) > 0$.

For the rest of the proof, we assume that $\|\alpha\|_\infty \leq n^{10^6 K}$. We will show that the total loss corresponding to $M_1$ is lower bounded by $n \cdot l(0) - o(1)$. This is sufficient since we already showed that the two copies of $M_2$ contribute a loss of at least $2n \cdot l(0)$.

Since all entries of the matrix $M_1$ are inversely superpolynomial (there is no pair of orthogonal vectors), we have that $|(M_1\alpha)_i| \leq n^{-\omega(1)}$ for all $i = 1, \ldots, n$. Using the second property again, the loss corresponding to $M_1$ is lower bounded by

$$n \cdot l(\pm n^{-\omega(1)}) \geq n \cdot (l(0) - o(1/n))$$
$$\geq n \cdot l(0) - o(1)$$

as required. □

# H  Gradient computation

We start from the proof of Lemma 15.

*Proof.*

$$\frac{\partial l_{\alpha,A,B}}{\partial \alpha_j} = \sum_{a \in A} \frac{\partial l(F_{\alpha,B}(a))}{\partial F_{\alpha,B}(a)} S(a, b_j) = l'(0) \cdot \sum_{a \in A} S(a, b_j) \qquad \text{(since } F_{\alpha,B}(a) = 0\text{)}.$$

□

**Sigmoid activation function**   We can show our hardness result holds also for the sigmoid activation function.

**Theorem 29.** *Consider a neural net with of size $n$ with the sigmoid activation function $\sigma(x) = \frac{1}{1+e^{-x}}$. Approximating the $\ell_p$ norm (for any $p \geq 1$) of the gradient of the empirical risk for $m$ examples within the multiplicative factor of $n^C$ for any constant $C > 0$ takes at least $O\left((nm)^{1-o(1)}\right)$ time assuming SETH.*

*Proof.* We set $S(a,b) := \sigma(-10(C+1)(\log n) \cdot a^\mathsf{T} b)$. Using Lemma 15, we get that the $\ell_1$ norm of the gradient is equal to $|l'(0)| \cdot \sum_{a \in A, b \in B} \frac{1}{1+e^{10(C+1)(\log n) \cdot a^\mathsf{T} b}}$. It is easy to show that this quantity is at least $|l'(0)|/2$ if there is an orthogonal pair and at most $|l'(0)|/(2n^C)$ otherwise. Since $l'(0) \neq 0$, we get the required approximation hardness. $\qquad\square$

**Polynomial activation function**   On the other hand, by using the polynomial lifting technique, we can show that changing the activation function can lead to non-trivially faster algorithms:

**Theorem 30.** *Consider a neural net with one hidden layer of size $n$, with the polynomial activation function $\sigma(x) = x^r$ for some integer $r \geq 2$. Computing the gradients of the empirical loss function for $m$ examples in $\mathbb{R}^d$ can be done in time $O((n+m)d^r)$.*

Note that the running time of the "standard" back-propagation algorithm is $O(dnm)$ for networks with this architecture. Thus our algorithm improves over back-propagation for a non-trivial range of parameters, especially for quadratic activation function when $r = 2$.

We start by defining the network architecture more formally. We consider a neural network computing a function $f : \mathbb{R}^{1 \times d} \to \mathbb{R}$ defined as $f(x) := S(xA)\alpha$, where

- $x \in \mathbb{R}^{1 \times d}$ is an input row vector of dimensionality $d$.
- $A \in \mathbb{R}^{d \times m}$ is a matrix with $j$-th column specifying weights of edges connecting the input units with the $j$-th hidden unit.
- $S : \mathbb{R} \to \mathbb{R}$ is a non-linearity that is applied entry-wise. $S(x) = x^r$ for some constant integer $r \geq 2$ for the rest of the section.
- $\alpha \in \mathbb{R}^m$ is column vector with $\alpha_j$ specifying the weight of the edge that connects the $j$-th hidden unit with the output linear unit.

Let $X \in \mathbb{R}^{n \times d}$ be the matrix specifying $n$ inputs vectors. The $i$-th row of $X$ specifies the $i$-th input vector. Let $z := f(X) \in \mathbb{R}^n$ be the output of function $f$ when evaluated on the input matrix $X$. Let $l : \mathbb{R}^n \to \mathbb{R}$ be the total loss function defined as $l(z) := \sum_{i=1}^n l_i(z_i)$ for some functions $l_i : \mathbb{R} \to \mathbb{R}$.

Let

$$\frac{\partial l}{\partial \alpha} := \left( \frac{\partial l}{\partial \alpha_1}, \ldots, \frac{\partial l}{\partial \alpha_m} \right)^\mathsf{T} \in \mathbb{R}^m$$

be the vector of gradients for weights $\alpha_1, \ldots, \alpha_m$. Let $\frac{\partial l}{\partial A} \in \mathbb{R}^{d \times m}$ be the matrix that specifies gradient of $l$ with respect to entries $A_{k,j}$. That is,

$$\left( \frac{\partial l}{\partial A} \right)_{k,j} := \frac{\partial l}{\partial A_{k,j}}$$

for $k = 1, \ldots, d$ and $j = 1, \ldots, m$.

**Theorem 31.** *We can evaluate $\frac{\partial l}{\partial \alpha}$ and $\frac{\partial l}{\partial A}$ in $O((n+m)d^r)$.*

*Proof.* Let $l'(z) := \left( \frac{\partial l_1}{\partial z_1}, \ldots, \frac{\partial l_n}{\partial z_n} \right) \in \mathbb{R}^n$ denote the vector that collects all $\frac{\partial l_i}{\partial z_i}$.

We note that

$$\frac{\partial l_i}{\partial \alpha_j} = \frac{\partial l_i}{\partial z_i} \cdot (\text{output of the } j\text{-th hidden unit on the } i\text{-th input vector})$$

$$= l'_i(z_i) \cdot (X^{(r)} A^{(r)})_{i,j}.$$

This gives

$$\frac{\partial l}{\partial \alpha} = \left( X^{(r)} A^{(r)} \right)^{\mathsf{T}} l'(z)$$

$$= \left( A^{(r)} \right)^{\mathsf{T}} \left( \left( X^{(r)} \right)^{\mathsf{T}} l'(z) \right).$$

The last expression can be evaluated in the required $O((n+m)d^r) = (n+m)^{1+o(1)}$ time.

We note that

$$\frac{\partial l}{\partial A_{k,j}} = \sum_{i=1}^{n} \frac{\partial l_i}{\partial A_{k,j}}$$

$$= \sum_{i=1}^{n} X_{i,k} \cdot r \cdot (\text{input to the } j\text{-th hidden unit})^{r-1} \cdot \alpha_j \cdot l_i'(z_i).$$

For two matrices $A$ and $B$ of equal size let $A \circ B$ be the entry-wise product. We define the column vector $v_k \in \mathbb{R}^n$: $(v_k)_i = X_{i,k} \cdot r \cdot l_i'(z_i)$ for $k = 1, \ldots, d$. Then the $k$-th row of $\frac{\partial l}{\partial A}$ is equal to $(v_k^{\mathsf{T}} X^{(r-1)} A^{(r-1)}) \circ \alpha^{\mathsf{T}}$. We observe that we can compute

$$(v_k^{\mathsf{T}} X^{(r-1)} A^{(r-1)}) \circ \alpha^{\mathsf{T}} = ((v_k^{\mathsf{T}} X^{(r-1)}) A^{(r-1)}) \circ \alpha^{\mathsf{T}}$$

in $O((n+m)d^{r-1})$ time. Since we have to do that for every $k = 1, \ldots, d$, the stated runtime follows. $\qquad\square$

## Footnotes

[9]For all $i \neq j$ we have $\bar{b}_i^\mathsf{T} b_j \geq 1$. This holds because all vectors $b_i$ are distinct and have the same number of 1s.