[Reviews · NeurIPS 2017]

Reviewer 1



The paper make use of (relatively) recent advances in complexity theory to show that of many common learning problems do not allow subquadratic time learning algorithms (given the veracity of the "Strong Exponential Time Hypothesis"). I appreciate that the authors do not oversell their results: They clearly state that they provide a worst-case analysis. Also, the results are not surprising. For instance, finding the exact solution of any kernel method requires the computation of the full kernel matrix, which is already quadratic in number of training examples. Reducing this computation time would imply that one can compute an approximation of the exact solution without computing the full kernel matrix, which is intuitively unlikely, unless he makes extra assumptions on the problem structure (i.e., the nature of the data-generating distribution). Hence, I would like the authors to comment on whether or not similar results could be obtained for the linear counterparts of studied ERM algorithms. That being said, I consider that this is an important contribution. Up to my knowledge, this work is significantly different than common statistical learning theory papers. It may bring fresh air to our community, by bringing tools and insights from the complexity theory literature. From my point of view, the reductions from a discrete optimization problem to a continuous one are clever. It is also worth noting that the paper is written rigorously (more than the average NIPS paper, I would say). Despite the hardness of the subject, it is easy to get the main ideas, notably the reductions from one problem to the other.

Reviewer 2



This is a theoretical paper arguing that the computation complexity of some important machine learning algorithms (such as SVM and deep learning) cannot be solved in less that sub-quadratic time. This paper is a short version of a 27 pages long paper published on Archiv. Unfortunately it contains references to the long paper (such as Sections B, C and D line 127, Section F line 137, Section E line 149, Section H lie 187, section B line 240, Section G line 277 and See Section H for other results on the last line) so that it is no longer self content and hardly readable for such a new result regarding the NIPS community.

Reviewer 3



This paper proposed a theoretical analysis for the fine-grained computational complexity of empirical risk minimiztion based on the Strong Exponential Hypothesis. Kernel SVM, Kernel Ridge Regression,Optimizing the last layer of NN are chosen as examples for convex cases and computing the gradient are chosen as an example for non-convex optimization problem. The conclusion is that no algorithm solve the aforementioned problems to high accuracies in sub-quadratic time. The proof is conducted by reducing OVP and BHCP problems to ERM problems and based on the fact that OVP and BHCP problems need quadratic time, authors inferred that ERM problems require almost quadratic time assuming SETH is true. My biggest concern is the way how authors infer the fine grained complexity of general ERM problems. OVP and BHCP can be treated as special cases of ERM. But are there some ERM problems that require sub-quadratic time. Because OVP and BHCP are just a subset of the ERM problems unless authors are discussing the worse case performance.

Reviewer 4



This paper provides an analysis of the time complexity of ERM algorithms. Specifically, the authors consider the ERM of kernel methods and neural network and give hardness results for computing the solutions of these problems. The main contribution is to show that these ERM problems require almost quadratic time if conjectures such as  the Strong Exponential Time Hypothesis (SETH) turns out to be true. Also, a similar result is given for computing the gradient in a neural network architecture. This paper is well written and clear. The subject matter of the paper is of importance, since understanding the computational complexity of solving ERM problems is a key issue to design new efficient learning algorithms. In this sense, this paper provides interesting new insights on the computational limits of ERM algorithms. The computational hardness results presented in this paper are interesting and novel to my knowledge. Although all these results are valid when assuming SETH, this provides an interesting common link between the two problems of ERM and satisfiability.